# A Study of Cumulative COVID-19 Mortality Trends Associated with Ethnic-Racial Composition, Income Inequality, and Party Inclination among US Counties

**DOI:** 10.3390/ijerph192315803

**Published:** 2022-11-28

**Authors:** Tim F. Liao

**Affiliations:** Department of Sociology, University of Illinois at Urbana-Champaign, Urbana, IL 61801, USA; tfliao@illinois.edu

**Keywords:** COVID-19 mortality, ethnic-racial composition, income inequality, party inclination, public policy, US counties

## Abstract

This research analyzes the association between cumulative COVID-19 mortality and ethnic-racial composition, income inequality, and political party inclination across counties in the United States. The study extends prior research by taking a long view—examining cumulative mortality burdens over the first 900 days of the COVID-19 pandemic at five time points (via negative binomial models) and as trajectories of cumulative mortality trends (via growth curve models). The analysis shows that counties with a higher Republican vote share display a higher cumulative mortality, especially over longer periods of the pandemic. It also demonstrates that counties with a higher composition of ethnic-racial minorities, especially Blacks, bear a much higher cumulative mortality burden, and such an elevated burden would be even higher when a county has a higher level of income inequality. For counties with a higher proportion of Hispanic population, while the burden is lower than that for counties with a higher proportion of Blacks, the cumulative COVID-19 mortality burden still is elevated and compounded by income inequality, at any given time point during the pandemic.

## 1. Introduction

This cohort study investigates county-level mortality trajectories of coronavirus disease 2019 (COVID-19) in the United States, a country with the highest cumulative COVID-19 burden globally [1]. The primary focus here is on the trajectorial ethnic-racial and economic association with COVID-19 death burdens [2,3] up to the 900th day of the pandemic by following up on an earlier study reporting strong cross-sectional county-level association of ethnic-racial and income inequality with COVID-19 incidence and mortality rates for the first 200 days of the pandemic in the US [3]. In addition, researchers found association between counties with greater Republican inclinations and a higher level of COVID-19 mortality [4]. Thus, the current research differs from the previous research [3,4] in both the length of the pandemic under examination and the type of analysis by providing an additional growth curve analysis for studying the differential growth curves of cumulative COVID-19 mortality.

Note that some recent studies of the ethnic-racial or sociodemographic association with COVID-19 mortality as well as comparative studies of COVID-19 fatalities in the US and the EU also analyzed cumulative mortality in the US [5,6,7]. Local public health departments often correct and update previously released COVID-related data, thus making analyzing weekly data less accurate and less desirable. In that sense, cumulative mortality data tends to lead to more stable analytic results. Besides, if we are concerned with the mortality burden of the COVID-19 pandemic on communities and local regions, county-level cumulative mortality reflects such a burden much more closely, a main rationale for some recent studies on the topic [8,9].

Based on the prior literature, the present study aims to extend what has previously been done by analyzing COVID-19 cumulative mortality in the counties in the US with a long view. Akin to relevant prior research [3], county-level COVID-19 mortality data collected by county public health departments were merged with data from other sources for the current analysis. The analysis in this paper relies on a single cohort of the counties at the start of the COVID-19 pandemic in the US focusing on the association between ethnic-racial composition, income inequality, and political party inclination on the one hand and cumulative mortality on the other, with a set of important county-specific factors under control in the estimation of the above-mentioned association. Therefore, the current study goes beyond the earlier US county-level COVID-19 mortality research [3,4,8,9] in three important ways: First, it covers a much longer period of the pandemic up to the 900th day since the first case in the US; second, it breaks up the 900 days into five cumulative periods to examine of the stagewise cumulative burden of COVID-19 mortality on these counties vis à vis the three important factors of ethnic-racial composition, income inequality, and political party inclination; third, it analyzes these five cumulative periods as mortality trends in a single growth curve analysis to reveal the relationship between such trends and the three important structural factors.

## 2. Methods

### 2.1. Data

This cohort study supplemented the data from the 3,141 counties in the 50 states and for Washington, DC, available from the seven major sources reported in the previous study [3] with the cumulative death data for the 200th, 400th, 600th, 800th, and 900th day of the pandemic, where day 1 is when the first case in the US was confirmed on 22 January 2020, up to 10 July 2022 (the 900th day), obtained from the same source as before [10]. Higher vaccination coverage was found associated with lower rates of COVID-19 mortality in the US [11], suggesting the inclusion of vaccination rates as a time-varying covariate. However, although good vaccination coverage data exists for the earlier periods, the definition of vaccination coverage later during the pandemic complicated data availability when the two booster shots were distributed during the period under study. Here, a county’s presidential Republican vote share before the start of the pandemic was used for measuring political inclination because research found a strong association between US counties’ Republican voter percentages and their vaccination rates [12]. The analysis reported below did not include Rio Arriba County, New Mexico due to missing information on income inequality. The study followed the Strengthening the Reporting of Observational Studies in Epidemiology (STROBE) reporting guideline.

### 2.2. Analysis

The analysis focused on a single outcome variable—the number of cumulative deaths (per 100,000 population) in a county—and 14 covariates as defined in Table 1. The mortality rate outcome was first analyzed with multilevel negative binomial models with level-2 (counties nested in states) random effects and a log link at the five time points of the 200th, 400th, 600th, 800th, and 900th day of the COVID-19 pandemic in the US. For each of the five time periods, we also have a “Days since 1st case” variable, measuring the exposure time since the first local case as a control in the analysis. All these covariates were then included in a growth curve analysis to obtain an estimation of the association between the set of covariates and cumulative mortality growth trends in the country. In this latter analysis, the natural logarithm transformed outcome (with a unit constant added to all, to facilitate natural logarithm transformation) was analyzed with two linear growth curve models—with the first model with random intercepts (representing counties) only and the second, random intercepts and random slopes for time as measured by the number of days of the pandemic. The data analyzed were based on as many multiples of 100 days as possible before the completion of the study. This decision leaves the last time interval 100 days instead of 200 as for the previous ones. For growth curve modeling, however, this creates no problems because such models can estimate data with unequally spaced time points [13].

Note that although the previous study to which the current study follows up reported significant interaction effects between ethnic-racial composition and income inequality for the first 200 days of the pandemic [3], a preliminary analysis revealed that only one out of six such interaction estimates for the last three time periods was statistically significant at the 0.05 level. Therefore, the current analysis does not include interaction terms between ethnic-racial composition and income inequality in either the stagewise or the growth curve analysis. However, in a nonlinear analysis such as a growth curve model where the outcome is natural logarithm transformed, the relations between the explanatory variables and the outcome become multiplicative, thereby capturing interactions in a different way. Parameter estimates and estimated mortality rates both at the 900th day (using the last multilevel negative binomial model) *and* over time (using the second growth curve model) were computed for interpretation.

Therefore, this two-part analysis allows us to see the over-time trends of the association between ethnic-racial/economic inequality as well as political inclination and COVID-19 cumulative mortality in two ways—analyzing both the association of COVID-19 mortality with the key structural variables at the five milestone time points and modelling such association as a latent growth curve function. This way, the stagewise and the growth-curve approaches can complement each other.

## 3. Results

Let us first focus on the estimates of the four key structural variables (Table 2) with ethnic-racial and economic inequality and political choice in voting as the key variables. Whereas a county’s percent population voted Republican before the Pandemic did not appear to have significant association with COVID-19 mortality in the previous analysis of the first 200 days of the pandemic [3], as shown in the first model, it shows strong association with cumulative mortality over 400, 600, 800, and 900 days of the pandemic in the remaining four models.

The persistent association patterns of ethnic-racial and economic inequality with cumulative COVID-19 mortality at 900 days are more complex: First, the estimates of percent Black, percent Hispanic, and Gini index all showed a consistent strong effect over the duration of 900 days of the pandemic; while ethnic-racial associations declined a bit since the initial 200 days, they remained highly significant over the remainder of the pandemic up to the 900th day; economic inequality as measure by the Gini index, albeit also declined a bit after 200 days, remained strong throughout the remaining period of the pandemic. These patterns can be better understood by plotting the estimated cumulative COVID-19 mortality rates at the 900th day of the pandemic by ethnic-racial and income inequality, with all other covariates under control (Figure 1). Compared to the analysis of the first 200 days [3], Black composition displayed a significant association with mortality (the left panel in Figure 1). With 0% Blacks in a county, the estimated death rate was 263.88 [95% CI, 235.47–292.29] at Gini value of 35% and for a county with 70% Blacks, the estimated rate increased to 912.26 [95% CI, 748.13–1,076.38] at Gini value of 55%. Hispanic composition showed a clear association with estimated mortality: The estimated mortality rate for a county with no Hispanics was 285.02 [95% CI, 254.06–315.97] at Gini value of 35% and for a county with 70% Hispanics, it was 735.01 [95% CI, 589.43–880.59] at Gini value of 55%.

The focus in the second stage of the analysis of cumulative mortality is on the estimates of the four key structural variables, especially those of ethnic-racial and Gini income inequalities (Table 3). These estimates differ little between the two models; we thus rely on Model 2 for interpretation by concentrating on the four structural variables. At any given time point during the first 900 days of the COVID-19 pandemic in the US, a 1% increase in a county’s GOP vote would imply a 1.7% increase in cumulative mortality; a 1% increase in a county’s Black population would mean a 1.9% increase in cumulative mortality; a 1% increase in a county’s Hispanic population would suggest a 0.7% increase in cumulative mortality; and a 1% increase in a county’s Gini inequality index would lead to a 1.4% increase in cumulative mortality, other things being equal. The typical proportion of variance explained as reflected by the standard *R*^2^ statistic is not available for negative binomial models. Rather, the model χ^2^ represents how well a model fits the data, compared to an intercept-only model; in other words, how much explanatory power the covariates have collectively. Judged by the χ^2^s, these five models are all distinctly different from the intercept-only model at the 0.0001 level.

Similarly, both growth curve models in Table 3 fit the data extremely well, even better than the stagewise models in Table 2, judged by the χ^2^ statistics. While most estimates are similar in terms of statistical significance, those for state-level variables may not be. The models reported in Table 2 are multilevel negative binomial models with random intercepts while those presented in Table 3 are linear grown curve models with either only random intercepts or both random intercepts and random slopes. State-specific variables such as ACA or Term limit are in fact in direct competition in estimation with state-specific random intercepts in a multilevel negative binomial model whereas such variables are not in direct competition in estimation in a growth curve model where random intercepts represent counties instead of states as in the stagewise models.

Because the four structural variables in the second model (Table 3) all display a strong positive association, and all are significant at the 0.001 level, to better understand the patterns of association, let us once again visualize the results of the three main structural variables of percent Black, percent Hispanic, and the Gini index by estimating mortality rates for the four scenarios of 0% and 70% of an ethnic-racial group in a county at Gini income inequality of 35% and 55% and present the estimates in Figure 2.

Over the course of the pandemic up to its 900th day, a county with 70% Black residents and a high level of income inequality (Gini = 55%) increased mortality rate much faster than a county with no Black resident and with the same inequality, passing 500 per 100,000 by the 600th day instead of just over 200 per 100,000 by the same time for a county with no Black residents (the left panel). Income inequality intensified the increase of mortality. This intensification is captured by the multiplicative effects of the covariates involved in a linear growth curve model with a natural logarithm transformed outcome variable. For comparing with the estimates above, a low level of inequality at a Gini of 35% dampened mortality by about 600 per 100,000 by the 900th day for the county with 70% Blacks and the dampening effect is much smaller, though still over 100 per 100,000 for a county with no Blacks. A high percentage of Hispanic population also increased mortality over the course of the pandemic, albeit to a lesser degree than for Blacks, and the effects of income inequality are also smaller (the right panel), compared to those for Blacks. Note that the 95% CIs for the estimates are wider for higher ethnic-racial compositions due to their smaller subsample sizes, though still with sufficient separation between estimated trajectories.

## 4. Discussion

The finding about the importance of voting for Republicans in a county before the start of pandemic is consistent with the extant literature, where the county-level share of Trump votes was found highly correlated with less physical distancing for COVID-19 considerations [14], and one’s political party affiliation is related to mask wearing, with Republicans more reluctant to do so [15]. In addition, the intensified association between Republican party inclination and COVID-19 mortality over the course of the earlier periods of the pandemic up to 31 October 2021 found by an earlier study [16] is confirmed here for an additional period of more than seven months, as showed by the first-stage analysis of Table 2. Possible explanations include, in addition to mask-wearing practices mentioned above, differential vaccination rates: Counties with higher Republican votes tended to have a lower full vaccination rate [12,17], and those who preferred personal freedom over vaccination tended to participate in the protests again the Biden administration’s vaccine mandates between November 2021 and January 2022 when such protests swept across cities in the US [18]. These reasons together may account for the association between political orientation and cumulative COVID-19 mortality up to 900 days of the pandemic. Therefore, in the current study, a county’s percent of voting for the GOP is found consistently associated with higher mortality throughout, especially later periods of, the COVID-19 pandemic.

The findings in the results section about the association between ethnic-racial and economic inequality and cumulative COVID-19 mortality have important consequences and suggest that counties with higher proportions of minorities bore greater COVID-19 cumulative mortality burdens deep into the pandemic in the US. Counties with proportionally more Black and Hispanic residents may have experienced residential racial segregation and other forms of discrimination [8]. Thus, for counties with a higher composition of Black population, the COVID-19 mortality burden is particularly high, especially those counties that also have a higher income inequality. For counties with a higher proportion of Hispanic population, the mortality burden, while lower than that for counties with more Black residents, still is elevated. When economic inequality intersects with ethnic-racial inequality in health, the interplay of race and class must be considered because both lead to health disparities [19]. Beyond the COVID-19 pandemic, environmental health is also affected by the interplay between ethnic-racial and economic inequality: The harmful effect of air pollution on life expectancy is especially pronounced where there is a higher level of income inequality and a larger number of Black populations [20]. Both of these findings suggest a need for policy changes.

Therefore, actions must be taken by institutions to reconcile and repair the marginalization that perpetrated society over a century [21]. Public health departments must concentrate their efforts on counties with higher ethnic-racial compositions by providing greater vaccination and COVID-19 medical support. Income inequality deepens such ethnic-racial marginalization over the course of the pandemic. Potential policy solutions to mitigate economic inequality include providing cash and in-kind support, increasing minimum wage, expanding the Earned Income Tax Credit, and adopting a progressive tax policy [22].

Future research can go beyond the three obvious limitations of the current study. First, it can go beyond the pandemic duration analyzed in this study because the COVID-19 pandemic is on-going, with new variants emerging. Second, it can include important time-varying covariates such as vaccination coverages with the proportion of population having had a certain number of vaccine shots at a given point in time when such data becomes available. Third, with a longer duration covered as suggested by the first point, the level of pandemic intensity can also be included in the analysis, such as peaks, plateaus, and troughs of the pandemic, offering another important time-varying covariate for analyzing COVID-19 mortality.

## 5. Conclusions

This study reported a cohort analysis of the association between ethnic-racial and economic inequality as well as political inclination and cumulative COVID-19 mortality from the beginning of the pandemic up to the 900th day and found the association strong, statistically significant, and persistent over the duration of the 900 days. Policies in support of reducing income inequality are needed, especially for communities where larger proportions of ethnic-racial minority populations reside and income inequality is higher, so that exorbitant COVID-19 mortality burden on such populations can be alleviated.

## Figures and Tables

**Figure 1 ijerph-19-15803-f001:**
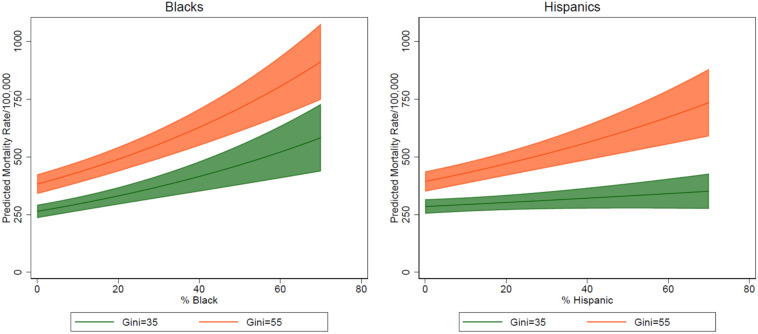
Cumulative Mortality Estimations at Day 900 by Ethic Composition and Income Inequality.

**Figure 2 ijerph-19-15803-f002:**
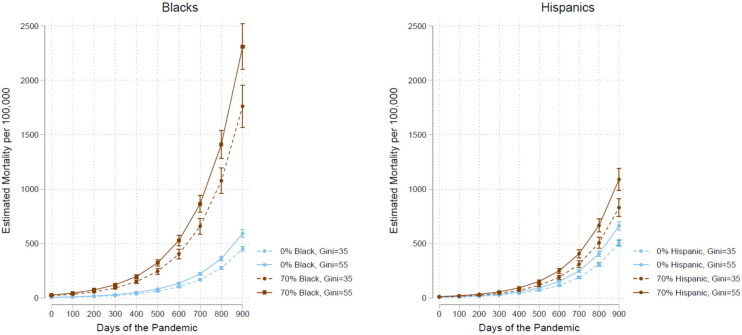
Mortality Estimations over 900 Days by Ethnic Composition and Income Inequality.

**Table 1 ijerph-19-15803-t001:** Descriptions of the Variables in the Analysis of 3,141 US Counties.

Variable	Definition	Mean (Range)	Source
Mortality 1	No. of deaths per 100,000 population, 1st period of 200 days	26.967 (0–413.858)	USAFacts.org; US Census Bureau
Mortality 2	No. of deaths per 100,000 population, 2nd period of 400 days	173.910 (0–865.801)	USAFacts.org; US Census Bureau
Mortality 3	No. of deaths per 100,000 population, 3rd period of 600 days	226.867 (0–865.801)	USAFacts.org; US Census Bureau
Mortality 4	No. of deaths per 100,000 population, 4th period of 800 days	358.021 (0–1.211.306)	USAFacts.org; US Census Bureau
Mortality 5	No. of deaths per 100,000 population, 5th period of 900 days	372.488 (0–2030.017)	USAFacts.org; US Census Bureau
% male	Percent male population, 2019	50.116 (42.992–73.486)	US Census Bureau
% Age < 20	Percent population under age 20, 2019	12.201 (0–22.443)	US Census Bureau
% Age ≥ 70	Percent population age 70 & over, 2019	6.751 (1.597–21.939)	US Census Bureau
ACA	States implemented Medicaid Expansion, 2020	0.547 (0–1)	USAFacts.org; US Census Bureau
Days since 1st case 1	Number of days for 1st period	71.169 (0, 200)	USAFacts.org; US Census Bureau
Days since 1st case 2	Number of days for 2nd period	71.762 (0, 400)	USAFacts.org; US Census Bureau
Days since 1st case 3	Number of days for 3rd period	72.023 (0, 600)	USAFacts.org; US Census Bureau
Days since 1st case 4	Number of days for 4th period	72.169 (0, 800)	USAFacts.org; US Census Bureau
Days since 1st case 5	Number of days for 5th period	72.247 (0, 900)	USAFacts.org; US Census Bureau
Population density	Population density per km^2^, 2019	105.495 (0.014–27,755.490)	US Census Bureau
% Black	Percent Black population, 2019	9.365 (0–86.593)	US Census Bureau
% Hispanic	Percent Hispanic population, 2019	9.754 (0.648–96.353)	US Census Bureau
Gini index	Gini index of income inequality	44.538 (25.670, 66.470)	2018 Am. Com. Survey
Term Limit	1 indicates yes	0.182 (0–1)	Council of State Governments
Governor Rep.	1 indicates Republican	0.569 (0–1)	National Governors Association
Governor male	1 indicates male	0.838 (0–1)	National Governors Association
Republican vote	2016 Republican vote, %	63.508 (4.122–95.273)	GitHub with 3 county-specific additions

**Table 2 ijerph-19-15803-t002:** Estimated Incidence RR From Multilevel Negative Binomial Models of Incidence in 3,141 US Counties.

	Model 1, 200 Days		Model 2, 400 Days		Model 3, 600 Days		Model 4, 800 Days		Model 5, 900 Days	
Covariate	RR (95% CI)	*p* Value	RR (95% CI)	*p* Value	RR (95% CI)	*p* Value	RR (95% CI)	*p* Value	RR (95% CI)	*p* Value
% male	1.007 (0.977–1.037)	=0.662	1.026 (1.015, 1.037)	<0.001	1.020 (1.010–1.029)	<0.001	1.018 (1.010–1.026)	<0.001	1.018 (1.009–1.026)	<0.001
% Age < 20	1.155 (1.091–1.222)	<0.001	1.120 (1.097, 1.144)	<0.001	1.110 (1.090–1.130)	<0.001	1.100 (1.083–1.116)	<0.001	1.097 (1.081–1.114)	<0.001
% Age ≥ 70	1.103 (1.042, 1.168)	<0.001	1.134 (1.110, 1.158)	<0.005	1.133 (1.113–1.154)	<0.001	1.130 (1.113–1.147)	<0.001	1.128 (1.111–1.145)	<0.001
Days 1st Case	1.014 (1.012, 1.017)	<0.001	1.000 (1.000–1.001)	=0.706	1.001 (1.000–1.002)	<0.001	1.001 (1.001–1.001)	<0.05	1.001 (1.001–1.003)	<0.01
Density	1.000 (1.000–1.000)	=0.828	1.000 (1.000–1.000)	<0.001	1.000 (1.000–1.000)	<0.001	1.000 (1.000–1.000)	<0.001	1.000 (1.000–1.000)	<0.001
ACA	1.054 (0.658–1.689)	=0.010	0.792 (0.590–1.100)	=0.173	0.806 (0.626–1.039)	=0.096	0.929 (0.745–1.158)	=0.512	0.925 (0.744–1.149)	=0.479
Term Limit	1.207 (0.733–1.987)	=0.460	0.954 (0.748–1.433)	=0.836	1.049 (0.805–1.368)	=0.724	0.955 (0.758–1.202)	=0.693	0.942 (0.751–1.181)	=0.605
Governor R	0.889 (0.592, 1.335)	=0.571	1.048 (0.670–1.147)	=0.336	0.874 (0.702–1.087)	=0.226	0.912 (0.754–1.103)	=0.340	0.903 (0.749–1.089)	=0.285
Governor M	1.134 (0.693–1.855)	=0.617	1.049 (0.629–1.202)	=0.398	0.889 (0.682–1.158)	=0.383	0.883 (0.701–1.111)	=0.289	0.889 (0.709–1.115)	=0.308
% GOP vote	0.999 (0.993, 1.005)	=0.777	1.005 (1.008–1.013)	<0.001	1.011 (1.009–1.013)	<0.001	1.013 (1.011–1.015)	<0.001	1.013 (1.012–1.015)	<0.001
% Black	1.026 (1.019–1.033)	<0.001	1.005 (1.012–1.017)	<0.001	1.014 (1.012–1.016)	<0.001	1.012 (1.010–1.014)	<0.001	1.012 (1.0100–1.014)	<0.001
% Hispanic	1.019 (1.013, 1.026)	<0.001	1.007 (1.007–1.012)	<0.001	1.008 (1.006–1.010)	<0.001	1.006 (1.005–1.008)	<0.001	1.006 (1.005–1.008)	<0.001
Gini index	1.031 (1.013, 1.049)	<0.001	1.012 (1.014–1.027)	<0.001	1.021 (1.016–1.027)	<0.001	1.020 (1.016–1.025)	<0.001	1.020 (1.015–1.025)	<0.001
var (state)	1.192 (1.227, 1.826)	<0.001	1.137 (1.116–1.323)	<0.001	1.137 (1.074–1.204)	<0.001	1.103 (1.014–1.035)	<0.001	1.099 (1.056–1.143)	<0.001
Model χ^2^ (*df*)	449.45 (13)		474.16 (13)		596.79 (13)		865.58 (13)		853.72 (13)	

**Table 3 ijerph-19-15803-t003:** Estimates from Growth Curve Models of Mortality Rates over 900 Days of the Pandemic in 3,141 US Counties.

	Model 1		Model 2
	Estimate (95% CI) *p* Value		Estimate (95% CI) *p* Value
Days in pandemic	0.005 (0.005–0.005) < 0.001		0.005 (0.005–0.005) < 0.001
% male	0.043 (0.035–0.051) < 0.001		0.030 (0.022–0.038) < 0.001
% Age < 20	0.112 (0.099–0.126) < 0.001		0.098 (0.084–0.112) < 0.001
% Age ≥ 70	0.177 (0.163–0.190) < 0.001		0.152 (0.138–0.166) < 0.001
Density	0.000 (0.000–0.000) < 0.001		0.000 (0.000–0.000) < 0.001
ACA	0.055 (0.015–0.096) < 0.01		0.050 (0.009–0.091) < 0.05
Term Limit	−0.049 (−0.094–−0.004) < 0.05		−0.063 (−0.109–−0.018) < 0.01
Governor R	0.028 (−0.007–0.062) = 0.115		0.015 (−0.020–0.049) = 0.410
Governor M	−0.170 (−0.212–−0.127) < 0.001		−0.171 (−0.214–0.127) < 0.001
% GOP vote	0.017 (0.016–0.019) < 0.001		0.017 (0.016–0.019) < 0.001
% Black	0.020 (0.019–0.021) < 0.001		0.019 (0.018–0.021) < 0.001
% Hispanic	0.007 (0.006–0.008) < 0.001		0.007 (0.006–0.008) < 0.001
Gini index	0.011 (0.007–0.016) < 0.001		0.014 (0.009–0.018) < 0.001
Constant	−4.484 (−5.069–−3.899) < 0.001		−3.592 (−4.185–−2.998) < 0.001
var(constant)	0.019 (0.012–0.032)		0.206 (0.185–0.228)
var(residual)	0.985 (0.964–1.008)		0.939 (0.919–0.960)
var(days)			2.83 × 10^−7^ (2.45 × 10^−7^–3.27 × 10^−7^)
cov(days, constant)			−0.00024 (−0.00026–−0.00022)
Model χ^2^ (*df*)	49,542.06 (13)		43,851.33 (13)
*N*	18,846 (=3,141 × 6)		18,846 (=3,141 × 6)

## Data Availability

The sources of the data used in this study are all cited in the references provided.

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
