# Peer review of "A Study of Cumulative COVID-19 Mortality Trends Associated with Ethnic-Racial Composition, Income Inequality, and Party Inclination among US Counties"

_ijerph, 2022, doi:10.3390/ijerph192315803_

Round 1

Reviewer 1 Report

The article addresses an important social issue that has been previously researched. The need for a new study is justified by the fact that retrospective data are more complete.

Abstract: "income mortality" - does not make sense. Did you mean "income inequality?"

Terms used: Define "ethnoracial" and "minoritized." What is "ethnicracial" and "ethnogracial"?

Explain unequal distances between chosen time periods, 200 days vs. 100 days. What motivated extending measurement to 900 days and not at 1,000 days (or stopping at 800 days)? 1,000 days would have been July 10, 2022. I want to see the data for this day to verify if the pattern holds. As a side note, Figure 2 shows consistent intervals of 100 days.

Method - explain "Days since 1st case 1-5" variables. The number of counties is stated twice in the same paragraph.

Results - What percent of variance in mortality do the significant predictors explain? If it cannot be calculated, explain why and provide evidence.

Why is Term Limit and Governor M significant in Table 3 and not in Table 2? What can account for these findings?

Discussion - Face mask wearing is also linked to party affiliation, an omitted explanation in this study, for example see https://onlinelibrary.wiley.com/doi/full/10.1002/ijop.12809

Conclusions need significant work. This section is about health equity. This term appears for the first time in conclusions and never mentioned elsewhere.

Author Response

Review 1:

The article addresses an important social issue that has been previously researched. The need for a new study is justified by the fact that retrospective data are more complete.

Thank you!

Abstract: "income mortality" - does not make sense. Did you mean "income inequality?"

That’s a typo. Indeed, it should be “income inequality,” and it’s now corrected.

Terms used: Define "ethnoracial" and "minoritized." What is "ethnicracial" and "ethnogracial"?

In the last several years, “ethnoracial” and “minoritized” have become more common in social science publications. However, I’ve changed them to “ethnic-racial” and “counties with higher proportions of minorities,” more common terms for a broader audience. I’ve also corrected the other typos that were variations of “ethnoracial.”

Explain unequal distances between chosen time periods, 200 days vs. 100 days. What motivated extending measurement to 900 days and not at 1,000 days (or stopping at 800 days)? 1,000 days would have been July 10, 2022. I want to see the data for this day to verify if the pattern holds. As a side note, Figure 2 shows consistent intervals of 100 days.

This is a great point. 1,000 days would be October 18, 2022, or 10 days after the submission of this manuscript. The motivation is to include as many multiples of 100 days as possible before the completion of the analysis, and the motivation did not impact the analysis negatively. This is now explained in a new passage with a new reference (p. 2), “The data analyzed were based on as many multiples of 100 days as possible before the completion of the study. This decision leaves the last time interval 100 days instead of 200 as for the previous ones. For growth curve modeling, however, this creates no problems because such models can estimate data with unequally spaced time points [9].” Figure 2 presents estimated mortality rates, and such estimations can be done for any time points.

Method - explain "Days since 1st case 1-5" variables. The number of counties is stated twice in the same paragraph.

An explanatory sentence is added near the reference to Table 1, “For each of the five time periods, we also have a ‘Days since 1st case’ variable, measuring the exposure time since the first local case as a control in the analysis.” The second mention of the number of counties is removed by rephrasing the sentence.

Results - What percent of variance in mortality do the significant predictors explain? If it cannot be calculated, explain why and provide evidence.

Thank you for bringing up this point. Conventional R2 statistic for proportion variance explained is not available for either negative binomial or growth curve models. Rather, we rely on the model χ2 statistic. This additional information is now provided for both Tables 2 and 3 while it was provided for only Table 2 in the previous version; additional interpretation based on χ2 statistics are now provided (p. 6).

Why is Term Limit and Governor M significant in Table 3 and not in Table 2? What can account for these findings?

This is a keen observation. The models reported in Table 2 are multilevel negative binomial models with random intercepts (representing the higher-level units—the states) while those presented in Table 3 are linear growth curve models with either only random intercepts (representing the counties) or random intercepts and random slopes (varying with time). State-specific variables such as ACA or Term limit are in fact in direct competition in estimation with state random intercepts whereas such variables are not in direct competition in estimation in a growth curve model where random intercepts represent counties. This is now clarified in a new paragraph after addressing the point above on p. 6.

Discussion - Face mask wearing is also linked to party affiliation, an omitted explanation in this study, for example see https://onlinelibrary.wiley.com/doi/full/10.1002/ijop.12809

Thank you for the suggestion, this omission is now filled with the suggested reference in the first paragraph of the discussion section (p. 7).

Conclusions need significant work. This section is about health equity. This term appears for the first time in conclusions and never mentioned elsewhere.

Apologies for this slip. I wrote the earlier version of the conclusion with a social scientist’s mindset but in the format for a health journal. This section is now rewritten.

Many thanks for the valuable comments!

Reviewer 2 Report

This research analyzes the association between cumulative COVID-19 mortality and ethnoracial composition as well as income inequality across counties in the United States. The study extends prior research by taking a long view—examining cumulative mortality burdens over the first 900 days of the COVID-19 pandemic at five time points (via negative binomial models) and as trajectories of cumulative mortality trends (via growth curve models). But there is still something that needs to be modified.

1.     Key words: The current six key words (COVID-19; Mortality; Health; Inequality; Race/Ethnicity; Income) do not summarize the core of this manuscript.

2.     In the introduction passage, specific data of the prior literature could be briefly described.

3.     In the methods part, because the study extends a long view(over the first 900 days), public health interventions(including vaccination and personal protection) in the United States also have dynamic changes,but the impact of this covariate was not fully considered. The author can consider revising this part.

4.     In the results part, the study revealed that a low level of inequality at a Gini of 35% dampened mortality by the 900th day for the county with 70% Blacks and the dampening effect is much smaller. A high percentage of Hispanic population also increased mortality over the course of the pandemic, albeit to a lesser degree than for Blacks, and the effects of income inequality are also smaller compared to those for Blacks. But there are interactions between income inequality or Gini index and ethnoracial composition. This part of the question needs to be discussed in more depth.

Author Response

Review 2:

This research analyzes the association between cumulative COVID-19 mortality and ethnoracial composition as well as income inequality across counties in the United States. The study extends prior research by taking a long view—examining cumulative mortality burdens over the first 900 days of the COVID-19 pandemic at five time points (via negative binomial models) and as trajectories of cumulative mortality trends (via growth curve models). But there is still something that needs to be modified.

Thank you! 

  1. Key words: The current six key words (COVID-19; Mortality; Health; Inequality; Race/Ethnicity; Income) do not summarize the core of this manuscript.

Good catch. The keywords are now changed to: COVID-19 mortality; ethnic-racial composition; income inequality; public policy; US counties

  1. In the introduction passage, specific data of the prior literature could be briefly described.

A sentence to that effect is now added in the introduction to highlight the county-level data analyzed by prior research.

  1. In the methods part, because the study extends a long view(over the first 900 days), public health interventions(including vaccination and personal protection) in the United States also have dynamic changes,but the impact of this covariate was not fully considered. The author can consider revising this part.

This is a great point. A detailed explanation/clarification is now provided in the data subsection of the methods section with two additional references (p. 2): “Higher vaccination coverage was found associated with lower rates of COVID-19 mortality in the US [9], suggesting the inclusion of vaccination rates as a time-varying covariate. However, although good vaccination coverage data exists for the earlier periods, the definition of vaccination coverage for the later three periods complicated data availability when the two booster shots were distributed. Here a county’s presidential Republican voting is used as a proxy because research found a strong association between US counties’ Republican voter percentages and their vaccination rates [10].

  1. In the results part, the study revealed that a low level of inequality at a Gini of 35% dampened mortality by the 900th day for the county with 70% Blacks and the dampening effect is much smaller. A high percentage of Hispanic population also increased mortality over the course of the pandemic, albeit to a lesser degree than for Blacks, and the effects of income inequality are also smaller compared to those for Blacks. But there are interactions between income inequality or Gini index and ethnoracial composition. This part of the question needs to be discussed in more depth.

This is a good observation. Indeed, the original study to which the current one is a follow-up found a significant interaction between Black composition and income inequality as well as between Hispanic composition and income inequality. However, in a preliminary analysis, only one of the six such interaction terms were found to be statistically significant at the 0.05 level for the last three time periods in the stagewise analysis. Therefore, interactions are not included as additional estimates in the final version of either part of the analysis; rather, interactions are only reflected in the nonlinear nature of the growth curve models because when the outcome variable is natural logarithm transformed, the effects between the explanatory variables on the outcome variable becomes multiplicative. It is now explained in three new sentences detailing all this in the methods section (p. 2). It is also clarified in an added sentence in the results section (p. 7).

Many thanks for your valuable comments!

Round 2

Reviewer 2 Report

The study aims to extend what has previously been done by analyzing COVID-19 cumulative mortality with a  long view. But contents with new focus and more innovative views needs to be added. So the readers may not be very interested.

Author Response

Thank you for the comment. This is a very good point. In the revised introduction, the last paragraph now reads:

“Based on the prior literature, the present study aims to extend what has previously been done by analyzing COVID-19 cumulative mortality in the counties in the US with a long view. Akin to relevant prior research [3], county-level COVID-19 mortality data collected by county public health departments were merged with data from other sources for this analysis with a long view. The analysis here relies on a single cohort of the counties at the start of the COVID-19 pandemic in the US focusing on the association between ethnic-racial composition, income inequality, and political party inclination on the one hand and cumulative mortality on the other, with a set of important county-specific factors under control in the estimation of the above-mentioned association. Therefore, the current study goes beyond the earlier US county-level COVID-19 mortality research [3,4,8,9] in three important ways: First, it covers a much longer period of the pandemic up to the 900th day since the first case in the US; second, it breaks up the 900 days into five cumulative periods to examine of the stagewise cumulative burden of COVID-19 mortality on these counties vis-à-vis the three important factors of ethnic-racial composition, income inequality, and political party inclination; third, it analyzes these five cumulative periods as mortality trends in a single growth curve analysis to reveal the relationship between such trends and the three important structural factors.”